# Geoscientists' views about science communication: predicting willingness to communicate geoscience

Joana Rodrigues[1], Cecília Castro [2], Elsa Costa e Silva [3], Diamantino Insua Pereira[1]

[1] Institute of Earth Sciences, Pole of the University of Minho, Braga, 4710-057, Portugal
[2] Center of Mathematics, University of Minho, Braga, 4710-057, Portugal
[3] Communication and Society Research Centre, University of Minho, Braga, 4710-057, Portugal

*Correspondence to*: Joana Rodrigues (joana225@gmail.com)

**Abstract.** The main barriers to science communication are common in different fields and they are widely identified in the literature. Studies focused on specific scientific communities framed science communication as an activity with the specificities of each context and field. In this study, we analysed geoscientists' representations and attitudes about communication to understand which factors can have significant impact on prediction of public engagement and that can explain the frequency/intensity of communication. The results pointed that factors such as professional experience, recognition by the institution, lack of financial support, personal satisfaction and geoscientific as an area of expertise, have a significant effect on their public engagement.

## 1 Introduction

Most scientists consider somehow important to communicate science (eg. Royal Society, 2006; Nielsen et al., 2007; Peters, 2013).It is not clear, however, whether this fact translates into effective communication (Jensen, 2011), as activities are mainly carried out as a 'goodwill exercise' (Neresini and Bucchi, 2011).

According to Burns et al (2003, p 191), science communication can be defined as the use of appropriate skills, media, activities and dialogue to produce personal responses to science, such as: awareness; enjoyment or other affective responses; interest, evidenced by voluntary engagement; opinions, forming, reforming and confirming science-related attitudes and understanding of contents, processes and social factors. It is considered an umbrella concept that includes several related terms such as 'dissemination', 'outreach', 'scientific literacy', 'popularization', 'informal education', 'public understanding of science', 'public awareness of science' or 'public engagement'. These concepts may have related goals but in fact, they are not synonymous and each one reflects a specific context. This is the case of 'public engagement', a term that refers to the current understanding of the relationship between citizens and science. A relationship that should be a dialogical and participative process (Bucchi and Trench, 2016), but that it is still not always the case. The type and nature of public engagement varies within specific communities and disciplines (Jensen and Croissant, 2007; Jensen, 2011; Kreimer et al., 2011; Johnson et al., 2014). Topics with greater popular visibility, public interest or increased 'social demand' such as medicine and health or climate, bring more attention to some areas than to others and give scientists different opportunities to engage with the public (Dunwoody et al., 2009; Jensen, 2011; Ivanova et al., 2013). For example, scientists who acknowledge the impact of their work tend to communicate more, such as nanoscientists, demonstrating greater sense of social responsibility (Dudo et al., 2014). Also, scientists who perceive the controversy of their research, as climate scientists (Entradas et al., 2019), or who understand the importance of communication for the well-being of society (Besley et al., 2013), feel a bigger responsibility to communicate. On the other hand, some scientists may feel that their disciplines have a less political nature and, therefore, see less value in public engagement, thus discouraging communication (Besley et al., 2013).

Generally, science communication intensity is higher in the social sciences and humanities areas (Kyvik, 2005; Jensen and Croissant, 2007; Torres-Albero et al., 2011; Jensen, 2011; Kreimer et al., 2011; Marchinkowski et al., 2014; Entradas and Bauer; 2017), despite natural sciences mobilising more practitioners in public events (Entradas and Bauer, 2017). Social Sciences and the Humanities engage more frequently with civic audiences and stakeholders, while Natural Sciences tend to address more often educational audiences (Entradas and Bauer, 2017).  It is possible that natural science and technology

scientists may engage less because lay public seem more interested in social, cultural or health topics (Bentley and Kyvik, 2011).

Natural and technology scientists experience more difficulties in making their research understandable when they communicate it (Bentley and Kyvik, 2011; Kyvik, 2005). Highly complex codes of the disciplines and difficulties of translation for non-experts are pointed out as barriers to communication (Bentley and Kyvik, 2011; Johnson et al., 2014). But, also, the behaviours and motivations differ: for example, astronomers participate regularly in engagement activities, what may be explained by astronomers' long tradition of outreach (Entradas and Bawer, 2019).

While previous studies showed that the majority of the scientific community, in general, was not involved in science communication (Jensen and Croissant, 2007), recent research presents evidence of higher levels of public engagement (eg. Rose et al., 2020). Also, specific communities such as climate scientists and astronomers have more often contact with lay audiences and are active communicators (Ivanova et al., 2013; Entradas et al., 2019, Entradas & Bauer, 2019). Jensen and Croissant (2007) followed the same community for several years and demonstrated that there is indeed an increasing trend in the number of activities. This trend may be mainly explained by external demands, related with the social impact of the topics

(Jensen, 2011; Ivanova et al., 2013). For example, scientists in Spain do not see the need to communicate because there is a little demand for it (Torres-Albero et al., 2011).

Literature reveals a wide range of studies focused on specific scientific communities like geneticists (Mathews et al, 2005), nanoscientists (Corley et al., 2011; Dudo et al, 2014), astronomers (Entradas & Bauer, 2019; Anjos et al., 2021), biologists and physicists (Ecklund et al., 2012, Johnson et al., 2013), climate scientists (Entradas et al., 2019) or marine scientists (Pinto

et al., 2018). A very first approach to study the geoscientists' community attitudes and practices was made by Liverman and Jaramillo (2011), under the scope of the Communicating Environmental Geosciences working group of the Commission for "Geoscience for Environmental Management" of the International Union of Geological Sciences (IUGS). That international study analysed environmental geoscientists' attitudes and experiences communicating science. The results of that study date back more than 10 years and comprise descriptive statistics, without advancing explanations about the reasons of these

behaviours.

Geoscience communication is an emergent area that still needs a better formalization, regarding its interdisciplinary approach and its specific challenges. It can be summarized as a practice which seeks to communicate aspects of geoscience with a wider audience, with the aim of increasing attention, involvement and public discussion of geoscientific results, aspects of outreach, public engagement, participation or knowledge exchange (Illingworth et al 2018).The main objective of this study is to maps

and to better understand Portuguese geoscience community communication attitudes in order to address their specific needs, providing clues for institutions, and policy makers to improve communication. Our specific goals are: to analyse geoscientists' representations and attitudes about science communication (1), to recognize the motivations and obstacles towards science communication (2), to understand which indicators explain geoscientists participation in public engagement (3), to identify the

factors that drive to higher participation in those activities (4) and to understand if the scientific nature/context of geosciences influences scientists' attitudes towards communication (5).

This study not only intends to assess the Portuguese geoscience community but also to bring wide inputs for the international community. These finds will contribute to develop a conceptual framework for geoscience communication research, identifying the main challenges and opportunities.

Research questions:

(RS 1) what explains geoscientists' public engagement?

(RS 2) what explains the intensity of this public engagement?

## 2 Background

Previous studies on scientific communities identified several predictors of public engagement (eg. Poliakoff and Webb, 2007; Dunwoody et al., 2009; Besley et al., 2013; Jensen 2011; Johnson et al., 2014), related with sociodemographic, context and personal factores.

### 2.1 Sociodemographic factors

Literature shows that sociodemographic variables influence science communication performance, but sometimes with inconsistent results. Some studies showed, for example, that women are more active (Andrews et al., 2005; Jensen, 2011; Ecklund et al., 2012; Jonshon et al., 2014) and give more importance to communication activities. Other studies demonstrated that men are more active (Crettaz von Roten, 2011) and they are more willing to communicate. More recent research concluded that gender does not affect (Entradas and Bawer, 2019). Also, age can affect participation in public engagement (Jensen, 2011), but together with gender and subfield, these factors seem to be considered minor predictors (Besley et al., 2013).

Formal training in science communication is considered a predictor (Dunwoody et al., 2009) and those who attends training are more critical of their own performance (Clarkson et al., 2018). But most scientists have no training (Royal Society, 2006; Entradas and Bawer, 2019; Ridgway et al., 2020) and geoscientists are no exception (Liverman 2009, Liverman and Jaramillo, 2011).

### 2.2 Context

Area of expertise

The contextual culture of each discipline influences the communication and type of activities (Johnson et al., 2014), as also researchers' opinion about their field and its impact on society. As we discussed earlier, the differences between communication practices of social and natural sciences and even between biology and physics have been studied. But when studying a community of geoscientists it seems also pertinent to try to understand if there are patterns or relationships between geosciences sub disciplines. The communication of geological hazards, an area with a large component of uncertainty and risk, and the geological resources, with a huge impact on society and local communities, or tectonics, a field whose content and practical applicability are far removed from the lay audiences, probably differ from each other.

For example, the abstract and not-human time scales of geoscience (see Bowring 2014) make public understanding difficult. On the other hand, environmental topics tend to be discussed in a "battlefield" mode, discouraging many scientists from engaging with the media (Boykoff, 2008). Also, many geoscience subjects are presented by the media mainly because of their disaster side, journalists rarely approaching positive aspects of research or direct implications on people's lives (Liverman and Jamarillo, 2011). Lack of strategic communication on topics with impacts on human health, for example regarding mining wastes, produced negative effects on local populations (Di Giulio et al. 2008). Not only the social impacts but also the political, cultural and economic orientations and the politicization of some subjects affects the will and way of communicating.

Public perception of their area of expertise

In general, scientists agree they have the moral duty to communicate, not only the results of their research but also social and ethical implications (Welcome Trust 2000, Royal Society 2006). According to Liverman & Jaramillo (2011), environmental geoscientists tend to consider that their work is not too complex or uninteresting for lay audiences. They also recognize the implications of their work for society and decision-makers and their moral duty to communicate. The subdiscipline is also related with the scientist's sense of responsibility to communicate and with the perceived public interest and complexity of

their area.

## Professional experience

The professional experience and career position may have positive effects (Jensen and Croissant, 2007; Jensen et al., 2008; Jensen, 2011), even if science communication activities are not officially recognized for career progression. Among academic

communities, senior researchers tend to be more active (Royal Society, 2006; Jensen, 2011; Kreimer et al., 2011; Torres Alberto, 2011; Entradas and Bawer 2019; Entradas et al., 2019). Also, the most academically productive researchers and with higher publication rankings engage more with public (Dunwoody and Ryan 1985; Jensen et al., 2008; Peters et al. 2008; Jensen, 2011; Bentley and Kyvik, 2011; Ivanova et al., 2013). These scientists tend to have more public visibility outside academic community (Jensen, 2011) and their status is positively correlated with the frequency of media engagement (Dunwoody et al.,

130    2009).

## Institution attitude

The context and attitudes of the institution influence the intensity of communication (Dunwoody and Ryan, 1985; Marchinkowski et al., 2014). Scientific institutions seem to be more and more committed with public communication and

performance indicators might prove their support, even so it is not considered an essential element (Neresini and Bucchi, 2011). Scientific careers and academic systems are still focused on research productivity and lack of encouragement and support from institution may be a demotivating factor for public engagement (Andrews et al., 2005; Ecklund et al., 2012; Shanley and Lopez, 2009; Rose et al., 2020). More recognition, rewards, encouragement from department heads would foster science communication activities (Royal Society, 2006).


## Barriers

Several barriers to public engagement have been pointed out by different scientific communities. Quite common are the lack of time (Pearson et al., 1997; Wellcome Trust, 2000; Andrews et al., 2005; Mathews et al., 2005; Royal Society, 2006; Nielsen et al., 2007; Poliakoff and Webb, 2007; Shanley and Lopez, 2009; Pinto et al., 2018; Ridgway et al., 2020), lack of financial

support (Poliakoff and Webb, 2007; Shanley and Lopez, 2009;, Pinto et al., 2018) or the lack of skills and training to communicate (Mathews et al., 2005; Shanley and Lopez, 2009), as discussed before.

The audience is also often identified as a barrier. Scientists see the public as a homogeneous group, irrational and misguided (Royal Society, 2006; Cook et al., 2004; Davies, 2008) and their seemingly lack of knowledge and interest discourages engagement (Wellcome Trust, 2000; Blok et al., 2008;, Besley and Tanner, 2011; Ecklund et al., 2012; Pinto et al., 2018;

Anjos et al., 2021), which is even more common in highly codified disciplines (as mathematics or chemistry) (Bentley and Kyvik, 2011).

In some cases, scientists feel that public engagement can cause negative opinion in their peers and have a negative impact in their career for making science too accessible and causing perceived reputational damages (Mathews et al., 2005; Royal Society, 2006; Jensen et al., 2008; Ecklund et al., 2011, Johnson et al., 2014).

The perception that scientists' visibility and frequency of media interaction might be inverse to their scientific ability and research is known as 'Sagan Effect', named after the astronomer and communicator Carl Sagan (Russo, 2010; Martinez-Conde, 2016). More recent studies show that peer critic opinions do not matter (Entradas and Bawer, 2018; Ridgway et al., 2020).

The relationship with the media has been always controversial (eg. Dunwoody and Ryan, 1985; Harz and Chappell, 1998; Nielsen et al., 2007; Corley et al., 2011; Anjos et al., 2021), even if the paradigm seems to be progressively changing (Peter et

al., 2008; Dunwoody et al., 2009; Ivanova et al., 2013). Scientists report critical opinion but positive experiences with journalists (Peters et al., 2008; Besley and Tanner, 2011), however they underline misrepresentation of scientific content by

journalists as an obstacle to communication (Blok et al., 2008; Young and Matthews, 2007; Hartz and Chappell, 1997; Mathews et al., 2005; Davies, 2008).

## 2.3 Personal factors

Self-perceived competence (self-efficacy)
Scientist's perception of their own skills and ability to communicate tends to have a positive impact on the intention to communicate and on their performance (Welcome Trust, 2000; Gascoigne and Metcalfe, 1997; Poliakoff and Webb 2007; Dunwoody et al., 2009; Ecklund et al., 2012). In general, scientists feel prepared to communicate (Welcome Trust, 2000; Royal Society, 2006), some even feel overconfidence (Rose et al., 2020). However, they feel less confident when it regards social and ethical implications (Royal Society, 2006).
The lack of communication skills and training can be considered a barrier to participate (Dunwoody and Ryan, 1985; Mathews et al., 2005; Poliakoff & Webb, 2007; Pinto et al., 2018) or not (Ridgway et al., 2020).

Personal satisfaction
Perception of personal performance may also provide indication about the attitude towards communication. Generally, scientists would like to spend more time with public engagement (Royal Society, 2006; Nielsen et al., 2007, Bentley and Kyvik, 2011; Liverman and Jaramillo, 2011), probably because it is not their main priority and they understand its benefits. Positive experiences and personal satisfaction is positively related to the intention to communicate (Andrews et al., 2005; Dunwoody et al., 2009; Entradas et al., 2019).

## 3. Data collection

To answer the research questions, (RS 1) what explains geoscientists' public engagement and (RS 2) what explains the intensity of this public engagement, a quantitative methodology was selected, with an attitude-based questionnaire, built for the purpose of this work considering the factors willingness and barriers to communication reviewed in the literature.

The data was collected through an online survey applied by self-administered questionnaire, through Google Forms, between June and July 2020, after a pilot test with 10 respondents.
The questionnaire consists of 47 questions (subdivided into 161 indicators) with an optional comment box. It is divided in three groups: demographic profile (a), experiences and practices (b) and representations (c) on science communication. It included closed-ended questions (multiple-choice, dichotomous, Likert scales, matrix, and ranking) and open-ended questions about geoscience topics and subjects. Since the survey applied is very extensive, with too many questions and indicators, we will focus on the present study on the data collected about the perceptions, analysing representations, beliefs, attitudes, preferences, motivations and expectations, in a total of 30 indicators.
An a-priori factorisation and a selection of the factors was performed under a strategic perspective drawn from the literature review and based on previous questionnaires such as Hartz and Chappell (1997), Wellcome Trust (2000), Royal Society (2006) and Liverman & Jaramillo (2011).
The complete questionnaire is available in the supplementary material A. The results referring to the 30 indicators studied in the scope of this work are also available in supplementary material D.

### 3.1. Data set

There is no official data on the number of geoscientists in Portugal, so a broad distribution of questionnaires was made by e-mail and social media. Scientific and professional associations and networks, research centres and university departments in Portugal were contacted to cooperate in the dissemination of the questionnaire among their members. It was also advertised in social media, in science communication and geoscience pages and groups. Despite the respondents being from different

sociodemographic groups, different professional categories and different fields of geosciences, the inference must be cautious since there is no guarantee of representativeness of the population under study. The data set consists on geoscience professionals and postgraduate students, developing their work in Portugal. A total of 179 valid responses was collected. As science communication practitioners are not only researchers and the academic community, data set also includes other professional. Among the surveyed geoscientists there were technical professionals (28%), university professors (20%), school teachers (18%), researchers (17%), postgraduate students (11%) and even science communicators (4%).

The surveyed geoscientists were 52% females, 56% were more than 41 years old, 40% have a PhD and 42% a Master degree. Regarding their degree area, the majority studied Geology (70%), followed by Biology and Geology / Environmental Sciences / Environmental Education (16%), and other areas such as Geological Engineering / Mine Engineering, Biology, Geophysics / Meteorology / Oceanography / Physics and Geography / Aerospace Engineering. The sample slightly overrepresented scientists with Geology degree (70%), with only 30% of geoscientists from the other degrees.

Regarding geographic distribution, responses were collected in all regions of Portugal, with a greater justifiable representation of areas with higher population density and greater concentration of scientific institutions (universities, research centres, science centres and Geoparks).

## 3.2 Methodology

Data was analysed by using SPSS, version 25.0 (IBM Statistics) and R, version 4.2.0. In addition to chi-square tests and Fisher's exact tests, used to test the independence between categorical variables, the statistical methodology consisted in the use of generalized linear models, with an explanatory objective (McCullagh and Nelder, 1989), since the main interest of this paper was to identify a set of features that can explain public engagement and the intensity of communication in science. In this work, we consider a result statistically significant if the p-value associated with the test statistic is $p < 0.05$ and statistically highly significant if the p-value is $p < 0.001$. Chi-square tests are statistical tests used to determine if there is a significant association between two categorical variables. The test is based on the difference between the expected frequencies and the observed frequencies in one or more categories.

The technique of multinomial logistic regression was used to model categorical response variables. This method is a type of regression analysis used for predicting a nominal dependent variable with more than two categories. It models the relationship between the independent variables and the dependent variable by estimating the probabilities of the outcomes for each category and choosing the one with the highest probability as the prediction. The model uses logistic regression for each of the categories and compares the results to determine the best fit. Through this model, factors affecting the class of the categories can be determined simultaneously. The multinomial response variable of intensity of public engagement consists of three categories: very active, active and inactive. An overview through the model is available as supplementary material B.

## 4. Results and discussion

Descriptive analysis of the results summarize the characteristics of the data set (supplementary material C). Regarding geoscientific areas, all the experts in External Geodynamics and Palaeontology and Geoconservation and Geotourism reported public engagement activities. Only 5.6% of the experts in Geological and Energy Resources did not report any activity. Also, 25% of the experts in History and Education or Environmental Geology and Environment and Engineering Geology reported no activity.

Concerning the institution attitude, 14,5% of respondents admit that their institution does not value communication activities, against 40,8% who consider that the institution gives some importance and 44,7% who say it that gives high value. These results show that the majority of the institutions have a perceived positive attitude towards communication that encourages geoscientist to engage with the public.

Concerning the perception about their geoscientific area, the majority (60%) does not think it is too complex to lay audiences and only 23% consider it to be too intricate. For, 36% this is an obstacle to communication. Most of them (79%) consider that

their area is interesting for non-experts and 85% admit that their work has implications for society and/or policy makers. The vast majority (90%) believes that scientists have a moral duty to engage with the non-expert public about the social and ethical implications of their work.

Regarding the obstacles to science communication, lack of financial support (87%) and lack of time (80%) are considered the biggest barriers. The majority (85%) does not think these activities make science less rigorous, 64% does not fear creating misunderstandings and generating controversy, but 75% believe that journalists' misrepresentation of scientific content is a barrier. This factor may partly explain, together with lack of opportunities, why the great majority of respondents reported none or rare contact with journalists or science journalists.

As for the audience's attitude, respondents believe that the public's lack of interest (64%) and lack of knowledge (61%) may be a constraint, but 64% does not see the complexity of their work area as an obstacle.

Regarding specific skills to communicate, only 56% agrees that lack of preparation/training can be an obstacle and 37% points out that discomfort in communicating with lay audiences can be a barrier. For 43% of the geoscientists, the negative opinion of their peers seems to somehow constrain.

Analysing the self-perceived competence, only 6% of the surveyed geoscientists admitted they do not have the necessary skills to communicate science. The vast majority feels prepared to communicate, with more than half (52%) being truly confident. Regarding more specific skills to communicate about the social and ethical implications of science, 25% recognized they do not feel prepared. At the same time, as seen above, almost half (44%) do not agree that lack of preparation or training may be an obstacle for effective communication.

Concerning the personal satisfaction, most of the respondents reported positive experiences and only around 1/4 feel dissatisfied. Almost half of the geoscientists (47%) think the number of communication activities they do annually is good and 12% think it is very good. However, an expressive amount of 40% thinks the number is reduced. For the vast majority of respondents (86%) it is gratifying to engage non-specialist audiences in science and 92% reported they find somehow important to find time to engage with non-expert audiences, demonstrating that geoscientists are aware of the importance of communication and they are interested in doing such activities.

**4.1 Factors associated geoscientists' public engagement**

In a first moment, we studied the influence that each isolated factor has on the response, regardless any of the others. Results from Chi-square tests (Table 1) demonstrated the relevant context and personal factors influencing the geoscientists' public engagement. No positive association was found with sociodemographic factors such as age or gender.

| Factors | Pearson Statistics and p-value | Conclusion |
|---|---|---|
| Area of expertise | X2 = 19.048, df = 10, p = 0.040 | Experts in Geological and Energy Resources tend to have more activity |
| Professional experience | X2 = 15.078, df = 6, p = 0.020 | Who has more than 20 years of experience tends to have more activity |
| Institution attitude | X2 = 26.135, df = 4, p < 0.001 | Who perceives more recognition tend to have more activity |
| Perceived implications for society | X2 = 12.636, df = 4, p = 0.013 | Who agrees more that its work has implications for society tends to have more activity |

| Lack of financial support | X2 = 12.886, df = 4, p = 0.012 | Who agrees more that lack of financial support is an obstacle tends to have more activity |
|---|---|---|
| Perceived complexity of their geoscientific subject | X2 = 11.920, df = 4, p = 0.018 | Who doesn't agree that the complexity of their geoscientific subject is an obstacle tends to have more activity |
| Self-perceived competence | X2 = 30.450, df = 4, p < 0.001 | Who feels more confidence in its skills tends to have more activity |
| Perception of the public engagement | X2 = 41.183, df = 4, p < 0.001 | Who thinks that performs a reasonable number of activities tends to have more activity |
| Personal satisfaction | X2 = 33.239, df = 4, < 0.001 | Who reports positive experiences tends to have more activity |

Table 1 - Results from chi-square tests

285 Regarding to RS 1 'what explains geoscientists' public engagement' we thus found that the factors area of expertise, professional experience, institution attitude, perceptions about the implications for society, lack of financial support, perceived complexity of their geoscientific subject the self-perceived competence, the perception of the public engagement and personal satisfaction, each have a significant effect on the response.

**4.2 Factors explaining the intensity of this public engagement**

290 To answer our RS 2, we used the approach of multinomial logistic analysis model that allows the selection of predictors that, together, have a significant effect on the dependent variable intensity of communication.

Participants were asked how many science communication activities had they carried out in the previous year. As no prior detailed explanation of 'science communication activities' was given, each respondent reported the activities they assume as such and which they consciously carry out as science communication.

295 The frequency/intensity variable was then organized into three categories: 'inactive' communicators, who reported not having carried out any activity in the last year (11.5%), 'active' that performed between 1 and 3 activities (35.3%) and 'very active' that reported more than 4 activities in the last year (53.2%). The reference class is 'very active'.

Multinomial logistic analysis began with the selection of important predictor variables using several selection feature techniques, including step-by-step forward and backward based on the log-likelihood of models. Among all possible predictors,
300 the best model (in terms of goodness of fit, based on the model's log-likelihood) includes the predictors: area of expertise, professional experience, institution attitude, lack of financial and personal satisfaction.
The goodness-of-fit indicator Nagelkerk's R2 value is 0.597. The confusion matrix (Table 2) shows that the overall percentage of correct ratings is 72.4%. Regarding the percentage of correct classifications, by category, the highest, 85.5%, is obtained in the category 'very active' and the lowest in the category, 'inactive' (55.6%). It is possible to conclude the model has a good
305 predictive ability. The coefficients table is available on the supplementary material B.

| *Classification* | | | | |
|---|---|---|---|---|
| | Predicted | | | |
| Observed | none | 1-3 | >4 | Percent Correct |
| none | 10 | 4 | 4 | 55,6% |
| 1-3 | 2 | 32 | 21 | 58,2% |
| >4 | 3 | 9 | 71 | 85,5% |
| Percent Correct | 9,6% | 28,8% | 61,5% | 72,4% |

Table 2 - Confusion matrix

Comparing 'inactive' with 'very active' (more than 4 activities), results show that the probability of not making any communication increases significantly in individuals with 5 to 10 years of experience, when compared to those with more than 20 years. The probability of engaging into public activities increases significantly in those whose institution appreciates their efforts. Therefore, if the institution values public engagement, the probability of an individual not making any communication decreases. For respondents with unsatisfactory experiences with public engagement, the probability of not making any communication increases. For geoscientists from areas of expertise like Geological and Energy Resources the probability of making activities increases.

Taking as reference the class 'very active' it is possible to conclude that the risk factors, i.e. increasing the probability of being 'inactive', are individuals with 5-10 years of experience and with unsatisfactory communication experiences. On the other hand, the probability of being 'active' (performing between 1–3 activities) increases for individuals those whose institution moderately or highly appreciates their efforts. The probability decreases for individuals with less than 5 years of experience and for those who do not see lack of financial support as an obstacle

Therefore, it is possible to conclude that having more than 20 years of experience, being in an institution who recognizes efforts in public engagement and working in Geological and Energy Resources areas fosters the increase of the number of communication activities per year, while previous unsatisfactory communication experiences are a risk factor.

For the RS 2 'what explains geoscientists' public engagement' we identified the following factors: the intensity of geoscientists' public engagement may be explained by the area of expertise, professional experience, institution attitude, lack of financial support and personal satisfaction.

Geoscientists from areas of expertise like Geological and Energy Resources, External Geodynamics and Palaeontology, Geoconservation and Geotourism are more likely to be active or very active in communication activities, than those from areas such as History and Education or Environmental Geology and Environment and Engineering Geology. On the other hand, the lack of support and the few years of experience are factors that may induce low frequency of science communication activities.

Despite the fact that, nowadays, scientific institutions seem to be more aware of the importance of public engagement, promoting and supporting initiatives and involving scientists, funding and career progressions still depend on research rankings. Many geoscientific institutions in Portugal promote the participation of their professionals in events such as Open Days, European Researchers Night, Living Science in Summer and at the School. Also, research projects funding requires work packages on the dissemination of results and outreach. However, all these endeavours are not considered on formal performance evaluation. Beside encouragement and financial resources, geoscientists could feel more committed to communicate with the public with formal recognition, logistical support, dedicated time and training.

On the other hand, we also concluded that less experienced professionals tend to engage less with the public. These results can be explained with the same argument of research rankings and lack of time, more crucial in early carriers. Also, the lack of public communication experience may deter geoscientists. Senior scientists with longer carriers, more recognized experience and visibility may also receive more demands.

As proved for other communities, we concluded that geoscientists with previous unsatisfactory experiences tend to communicate less. Lack of training, lack of support, non-expected impacts or so many other reasons can explain the dissatisfaction. This research focuses on scientists rather on audience, so we can only conclude about the intensity of public engagement and not about quality or impact of communication. A solution to overcome this barrier would be to follow closer these professionals, understand their difficulties and support them in overcoming them.

Regarding the area of expertise, despite dealing with complex and abstract processes and time scale, scientists from areas such External Geodynamics and Palaeontology tend to communicate more. Fossils (including dinosaurs) and landscapes may be difficult to understand but can be more appealing for non-expert audience.

Geoconservation and Geotourism experts work with geological heritage, an appealing topic and usually have public communication included in their duties. Some of them work in institutions like museums, science centres or geoparks and tend to be more motivated. Geological and Energy Resources are topics currently with high media visibility and social debates demanding geoscientists to engage more with public.

History and Education experts seem to engage less probably because these areas are extremely applied.

At last, geoscientists from areas such Environmental Geology and Environment and Engineering Geology tend to communicate less. This result could not have been expected, as these areas include topics such as climate change, soil and water contamination, topics with great social impact, on people's health and life. On the other hand, these areas have great visibility in media and public debate, discussions are very polarized and politicized and very often geoscientist face uncomfortable experiences. These results show that communication attitudes also depend on area of expertise. Interventions to foster public engagement among geoscientists should be target to different areas of expertise. Areas with more public visibility and polarization could benefit from targeted training, addressing the specific challenges they face and working together with social and human scientists. Training for geoscience graduation students, designed to answer specific challenges such as risk and emergency response, proved to be very effective improving communication skills (Dohaney et al., 2015).

Regarding the limitations of this experimental design, the sample selected may not be representative of the population being studied. Another limitation is the issue of imbalanced samples, as the distribution of the independent variable is not equal in each group of the experiment.

About the methods limitations, the small sample size needs to be considered. The multinomial logistic regression model requires a relatively large sample size in order to achieve accurate and stable estimates, particularly when there are multiple categories and many independent variables. The limited prediction accuracy is another limitation, since the multinomial logistic regression model may not always provide accurate predictions, especially when the relationships between the variables are complex and non-linear. Multinomial logistic regression is designed for categorical dependent variables and it assumes that the categories are nominal and have no inherent order or ranking.

## 5. Conclusions

The results of our study show that the majority of the surveyed geoscientists have positive impressions regarding public engagement, being motivated, felling comfortable and prepared to communicate. The main outcomes of the research confirm the patterns of scientists from other areas previously studied.

We concluded that public engagement may be explained by context factors such as area of expertise, professional experience, institution attitude, perceptions about the implications for society, lack of financial support and perceived complexity of their geoscientific subject and personal factors as the self-perceived competence, personal satisfaction and the perception of the public engagement.

Furthermore, we concluded that what really matters for the intensity of this public engagement are the context factors, like the area of expertise, personal experience, institution attitude, lack of financial support, and personal factors such as satisfaction. These should be the main factors to be considered when designing policies to support and promote public engagement, as they identify specific aspects that are more likely to lead to foster results.

The findings discussed above can support the development of strategies and recommendations that will contribute to overcome the constraints and lead to a more effective communication between scientists and society. Further research on geoscientist's representations would allow to understand the relation between perceptions and their practices.

Regarding the limitations of this research, we recognize a possible sampling bias. Firstly, it is a set of data that may not be representative of the entire population since the respondents were self-selected. Although the questionnaire was destined to both communicators and non-communicators, we assume the possibility that scientists less aware of communication may have responded less, leading to an overrepresentation of those who communicate. Also, in surveys like this in which people are asked to provide personal opinions and report personal practices, it is necessary to take into account that people may be led to answer according to socially desirable ideas. To try to reduce this bias and obtain more truthful answers, participants were informed that their identities were protected and the results would be appropriately anonymized. At the same time, online self-administered questionnaires, without the presence of other people, as it is the case, also contribute to reduce social desirability bias.

More robust results could be possible with a larger sample, for example analysing an international community or following the community for an extended period of time (e.g. Jensen & Croissant 2007). Also, complementary interviews could bring additional qualitative data. Despite these constrains, the sociodemographic heterogeneity of the data set, as well as the comparison of our outcomes with previous studies, reinforces the confidence in the results.


## Competing interests

The authors have declared that there are no competing interests.

## Ethical statement

The work performed in this study is original, reflects the authors' view and it does not engage in any form of malicious harm to other persons. The participation in the survey was voluntary and anonymous.

## Data availability

The survey and the statistical results are available in supplementary material.


## Author contribution

JR, ES, DP planned the research; JR collected the data; CC and JR analysed the data, JR prepared the manuscript draft, JR, CC, ES, DP contributed to the writing, review, and editing.

## Acknowledgements

Authors would like to thank the referees for their careful reading and for their constructive suggestions.
This work is supported by national funding awarded by FCT - Foundation for Science and Technology, I.P., projects UIDB/04683/2020 and UIDP/04683/2020.

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
