# Peer review of "Geoscientists' views about science communication: predicting willingness to communicate geoscience"

_EGUsphere, 2022_

## Author Response (AR1)

Authors' responses:

**Referee 1**

We sincerely thank Referee Rolf Hut for the valuable suggestions and constructive comments, which we found extremely useful to improve our manuscript.

1. On the survey design

Undoubtedly, we consider interaction with journalists as 'science communication activities' and the survey includes specific questions about journalists and media, such as Q43, Q17, Q37.11 (not analysed under the scope of the present study, with the exception of Q37.11). In Q11 were listed activities carried out directly with the 'broad public' and nonspecific for particular targeted publics, such as the media. Furthermore in Q13 the respondents reported the groups with whom they have engaged, where journalists were included. We will include in the appendix table the analysis of this question assessing the frequency of contact with journalists (Q13.1, Q13.2), where it is possible to conclude that it is very small.

Participants were not given any definition of 'science communication activities' in the survey. We did not intend to condition each personal notion/perception of this activities. Each respondent reported the activities they assume as such and which they consciously carry out as 'science communication'.

We agree that it may be relevant to add this circumstance in the text in section 4.2, before line 270:

To access this variable participants were asked how many science communication activities had they carried out in the previous year. As no prior detailed explanation of 'science communication activities' was given, each respondent reported the activities they assume as such and which they consciously carry out as science communication.

2. On the methodology

It was not conducted a sensitivity analysis in this study. We agree that it is important to mention in the text, on section 3, that when designing the questionnaire we did a-priori factorisation and the choice was made from a strategic point of view, considering the literature review, based on previous questionnaires such as Hartz and Chappell (1997), Wellcome Trust (2000), Royal Society (2006) and Liverman & Jaramillo (2011).

3. On the interpretation of the results

In section 3.1 (lines 193 - 197) are listed the geoscientists' degree area (Q4): Geology; Biology and Geology / Environmental Sciences / Environmental Education; Geological Engineering / Mine Engineering; Biology; Geophysics/ Meteorology / Oceanography / Physics and Geography / Aerospace Engineering.

The early training (Bachelor's) area does not always correspond to the area of work and research, therefore area of expertise was introduced as a different indicator (Q5): Geological and Energy Resources; Internal Geodynamics, Geophysics, Petrology and Geochemistry; External Geodynamics and Palaeontology; Geoconservation and

Geotourism; History and Education; Environment, Environmental Geology and Engineering Geology.

This detailed information will be included, in supplementary material, in the summary of the results of the demographic and descriptive analysis.

There is no data on the number of geoscientists in Portugal, but considering that the geosciences include several areas, besides geology, and that 70% of the respondents are geologists, we believe that the proportion of the other geoscientists is over represented.

Geology (125 respondents)

Biology and Geology/Environmental Sciences/Environmental Education (28 respondents)

Geological Engineering/Mine Engineering (8 respondents)

Biology (8 respondents)

Geophysics/Meteorology/Oceanography/Physics (6 respondents)

Geography / Aerospace Engineering (4 respondents)

This detailed information will be included, in supplementary material, in the summary of the results of the demographic and descriptive analysis.

We agree that a table that compiles the results of the descriptive analysis provides an easier and better overview (lines 211 – 243), clarifying for example the two previous discussed topics. We will submit it as supplementary material.

We also agree that for questions where high correlation between answers are important to understand it is relevant to include an overview of these correlations. As it is a significant amount of information, we will submit it as supplementary material.

We are aware that a social desirability bias always exists in these studies and that is also why we mention in the conclusions that more robust results may be obtained through further studies.

We agree that social desirability bias should be addressed in the text, around line 361:

In surveys like this where people are asked to provide personal opinions and report personal practices it is necessary to take into account that people may be led to answer according socially desirable ideas. To try to reduce this bias and obtain more truthful answers, participants were informed that their identities were protected and the results would be appropriately anonymized. At the same time, online self-administered questionnaires, without the presence of other people, as this case, also contribute to reduce social desirability bias.

4. On Open Science best practices

We will include in section 3, together with the methodology, the information regarding the software used for statistical analysis.

Under this research project we are currently still preparing the publication of other studies with all these unpublished data as also the submission of the full data collected with this survey to a specialised journal for open research data. For this reason, we consider to submit only the variables analysed in the scope of this study.

Once again, we thank Referee Rolf Hut for the constructive insights. We hope that our answers have clarified all the comments.

The authors
* * *
**Referee 2**

We thank Referee 2 for the constructive and helpful comments, which were very insightful to improve our manuscript.

Major comments:

1. We are aware that the samples of the different professional groups are relatively small and may not represent accurately their populations. Because of that we haven't processed their results separately. As the aim of the current work is to study the population in general, this question was included precisely to show that it is not a homogeneous population and that geoscience communication practitioners are not only researchers and academics.

Suggested by the Reviewer 1, we will include as supplementary material an overview through the results and correlations that will also present the professional groups distribution.

2. The expected frequency was calculated based on the assumption of independence between the two categorical variables being analysed and the chi-square test of independence was used to determine if there is a significant association between the two variables.

The analysis of individual factors was done by performing several independent chi-square tests to examine the relationship between the response variable and each predictor variable. This is a common approach for understanding how different factors may affect the response, as the reviewer well points out. However, as is well known, this approach has several limitations such as leading to an increased risk of Type I errors (i.e., false positives), because each test is evaluated separately, and the probability of making a Type I error increases with the number of tests performed. Furthermore, this approach does not allow for the analysis of multiple predictor variables simultaneously, so it is difficult to determine the relative importance of each predictor variable or to understand the combined effect of multiple predictor variables

on the response. To deal with these well-known limitations, we decided to also use a multinomial logistic regression model, a method that seems to us to be more appropriate, as it allows the analysis of multiple predictor variables simultaneously and provides estimates of the relative odds of each level of the response variable occurring. This can help to better understand the relationship between the predictor variables and the response variable, and to identify the factors that are most influential in determining the response.

3. Both Structure Equation Model, SEM, and ordinal regression are statistical techniques that can be used to analyse relationships between predictor variables and an ordinal response variable. SEM is a more comprehensive model that allows for the analysis of multiple mediator and moderator variables, but this is not exactly the purpose of this study. On the other hand, ordinal regression models are primarily concerned with estimating the predictor and response variables, which is the aim of this study.

However, we chose to use a multinomial logistic regression model which has the advantage of having been specifically designed for analyzing relationships between predictor variable and a categorical response variable with more that two levels. We think it is a good choice where the research question involves examining the influence of predictor variables on the likelihood of different levels of the response variable occurring.

Another advantage of multinomial logistic regression is the simplicity of interpretation. It involves fitting separate logistic regression models for each level of the response variable, and the coefficients from these models can be used to estimate the relative probabilities of each level of the response occurring.

There are other reasons that led us to choose this procedure:

1) the goal is to identify the predictors of membership in the different categories of the response variable;
2) the categories of the response variable are not equally spaced, and ordinal regression assumes that the distance between levels of the response variable is equal;
3) the sample size is small, so a multinomial logistic regression model may provide more stable estimators.

Minor Comments:

- We will revise the introduction part to make it more readable.

- Indeed, public engagement and science communication are not synonymous. Science communication refers to a broad range of activities, but given that, in the current paradigm of science communication, scientists communicating are mostly required to promote public engagement, these terms are normally used interchangeably, even in the specific literature regarding science communication. But we agree that this may not be understood as such by everyone so we consider it to be pertinent to clarify these concepts, and we will do so in the section 1 Introduction

- The studies on this specific topic are very few, almost non-existent among international literature, with one exception made more than 10 years ago (Liverman & Jaramillo, 2011). At the same time, the main outcomes of the research confirm the patterns of scientists from other areas previously studied. We think that a more general title may boost more opportunities for further researches to develop strategies to suppress geoscience communication constraints worldwide.

- We agree that it is pertinent to include in the introduction a more precise definition about geoscience communication and its scope within this study.

Once again, we thank Referee 2 for the relevant questions raised which we hope we have fully answered.

The authors

---

## Referee Report (RR1)

This paper is an improvement from the version that was submitted last time. Significantly the supportive materials have primarily enriched with statistical facts and explanations of analysis procedures and methods. The data enhanced the most valuable part of this paper: the extensive and detailed survey about factors affecting geoscience communication and public engagement. The supplementary explanation in the manuscript and the justification of the methods are reasonable.  However, this paper still needs minor changes.

Major comments:

In the methodology part, the authors only added the software and very technical data and SPSS and R data analysis procedure. Considering the Geoscience Communication journal's readers are mostly geoscience majors or communicators. I doubt the current methodology is useful for most of these readers. I recommend the authors give a bit of a simple explanation of the selected methods, even though they are standard methods for analyzing discrete data. This will also benefit other geoscientists interested in quantitatively analyzing similar surveys or designing similar experiments.

Secondly, to be informative, adding more information about the limitations of the methods and the experiment design would be appreciated mainly so that people can understand to what extent they can apply the insights to their situations. This limitation section can be a single section that inserts between the discussion and conclusion or add to the discussion.

Again overall, I think the study has good potential, and I look forward to reading the revised version.

---

## Author Response (AR2)

Authors' response:

We thank the Referee for the comments, in order to improve the quality of our manuscript.

As suggested, we included a simple explanation of the selected methods. We also added to the discussion some more information about the limitations of the methods and the experiment design.

Once again, we thank the Referee for the helpful insights.

The authors